# Ecological Aerobic Ammonia and Methane Oxidation Involved Key Metal Compounds, Fe and Cu

**DOI:** 10.3390/life12111806

**Published:** 2022-11-07

**Authors:** Hina Ayub, Min-Ju Kang, Adeel Farooq, Man-Young Jung

**Affiliations:** 1Interdisciplinary Graduate Programm in Advance Convergence Technology and Science, Jeju National University, 102 Jejudaehak-ro, Jeju 63243, Korea; 2Research Institute for Basic Sciences (RIBS), Jeju National University, 102 Jejudaehak-ro, Jeju 63243, Korea; 3Department of Science Education, Jeju National University, 102 Jejudaehak-ro, Jeju 63243, Korea

**Keywords:** ammonia and methane oxidation, monooxygenase, copper, iron, methanobactin, siderophore

## Abstract

Interactions between metals and microbes are critical in geomicrobiology and vital in microbial ecophysiological processes. Methane-oxidizing bacteria (MOB) and ammonia-oxidizing microorganisms (AOM) are key members in aerobic environments to start the C and N cycles. Ammonia and methane are firstly oxidized by copper-binding metalloproteins, monooxygenases, and diverse iron and copper-containing enzymes that contribute to electron transportation in the energy gain pathway, which is evolutionally connected between MOB and AOM. In this review, we summarized recently updated insight into the diverse physiological pathway of aerobic ammonia and methane oxidation of different MOB and AOM groups and compared the metabolic diversity mediated by different metalloenzymes. The elevation of iron and copper concentrations in ecosystems would be critical in the activity and growth of MOB and AOM, the outcome of which can eventually influence the global C and N cycles. Therefore, we also described the impact of various concentrations of metal compounds on the physiology of MOB and AOM. This review study could give a fundamental strategy to control MOB and AOM in diverse ecosystems because they are significantly related to climate change, eutrophication, and the remediation of contaminated sites for detoxifying pollutants.

## 1. Introduction

Nitrogen is an essential element for all living life on our planet, as it is a component of nucleic acids and proteins and constitutes most of the atmosphere, around 80%. The nitrogen in the ecosystem is cycled by various biological processes such as nitrogen fixation, nitrification, denitrification, assimilation, and ammonification. These processes include anaerobic nitrate reduction to ammonium (DNRA), denitrification of anaerobic methane oxidation (DAMO), and dissimilatory nitrate reduction to ammonium (DNRA) [1]. Nitrification is a vital process of the global biogeochemical nitrogen cycle. It plays a significant role in fertilizer loss in industrial agriculture, eutrophication, and the production of greenhouse gas N_2_O, which has a very long residence time in the atmosphere (120 years). Furthermore, it contributes to ozone destruction by reacting with the atomic oxygen to form nitric oxide (NO) in the atmosphere [2]. On the other hand, nitrification is essential for efficient sewage treatment. The aerobic oxidation of ammonia initiates nitrification by ammonia-oxidizing microorganisms (AOM), which is mediated by three distinct groups of aerobic autotrophic ammonia oxidizers: (i) ammonia-oxidizing bacteria (AOB), (ii) ammonia oxidizing-archaea (AOA), and (iii) complete ammonia-oxidizing bacteria (Comammox) [1]. Ammonia is also oxidized by the anammox (anaerobic ammonium oxidation) process in the anaerobic system [3] and also oxidized by the heterotrophic and fungal nitrification process [4]. However, these oxidation processes involve completely different physiological pathways compared to the canonical aerobic oxidation in AOM (see details below). Hence, in this review, we focused on canonical aerobic ammonia oxidation, which is physiologically comparable to the aerobic methane oxidation pathway.

Carbon dioxide (CO_2_) is the most prevalent greenhouse gas accounting for 95% of all emissions. The following two gases are methane (CH_4_) and nitrous oxide (N_2_O), which have a substantial impact on the climate [2]. Despite having atmospheric concentrations of ~1800 parts per billion (ppb) for CH_4_ and ~330 ppb for N_2_O [5], respectively, they are 25 and 300 times more effective at absorbing infrared light than CO_2_. Anaerobic decomposition of organic matter produces CH_4_ and later transforms it into CO_2_, increasing total atmospheric CO_2_ concentration. As a source of energy, methanotrophs (MOB; aerobic methane-oxidizing bacteria) are primarily responsible for the enzymatic oxidation of CH_4_. The aerobic and anaerobic methanotrophic reactions consume about 35% (0.6 Gt, gigatonne), and 18% (0.3 Gt) of the global CH_4_ production per year, respectively [6]. Canonical aerobic methane oxidation is performed by certain bacteria that combine molecular oxygen with CH_4_ to produce methanol (CH_3_OH), then the methanol is oxidized into formaldehyde (CH_2_O), and then finally oxidized to CO_2_ via formate (CH_2_O_2_) using either the serine or ribulose monophosphate pathway (Figure 1 and details see below) [7]. An anaerobic pathway of reverse methanogenesis is believed to oxidize methane by the ANME group of archaea [6,8], *Methanosarcinales* and *Methanomicrobiales*, which are closely associated with sulfate-reducing gamma-proteobacteria [9].

Interactions between microbes and trace metals such as copper (Cu), zinc (Zn), cobalt (Co), molybdenum (Mo), selenium (Se), manganese (Mn), and iron (Fe) are important in nature. Metals can influence microbial growth, activity, and survival directly (such as making metalloenzyme; see below) or indirectly [10]; thus, it is not unusual that they deal with them, sometimes to their use (bioavailable), often to their harm (toxic), when present at high enough concentrations [11]. Microorganisms can bind metal ions and transport them into the cells for various purposes, such as electron donors or acceptors in metabolic activities [12]. Most enzyme classes contain metalloenzymes, which are metal-bound proteins having a labile coordination site. The metal ion with a substrate-compatible shape is usually located in the active site of metalloenzymes. Therefore, as with all enzymes, the shape of the active site is crucial. Despite the intricacy of organic chemistry reactions, metal ions can execute such reactions that are challenging to achieve. The Irving-Williams stability series defines the order of affinity for essential divalent cations. In addition, the required concentration of trace metal can be calculated by the speciation of metals in solution, based on equilibrium stability constants (*K*) which is the strength of the interaction between metals and the ligands that come together to form the complex [13].

Therefore, the microbes must deal with enough metal atoms to satisfy the protein requirements; however, not all metals are bioavailable. In the environmental system, metals are found in various forms, including the hydrated free ion, inorganic complexes (with ligands such as Cl^−^, OH^−^, CO_3_^2−^), organic complexes (with simple organic molecules of biogenic or anthropogenic origin, and with natural organic matter, NOM), as well as in colloidal and other solid phase forms. It is well known that ligand concentration, temperature, pH, and redox state determine the partitioning (bioavailability and toxicity) of metals among the different forms [14,15]. In the natural environment, metals are usually present in complex and colloidal forms and are rarely found in free form [16]. The concentration of free metal ions can be decreased in the presence of chelating agents such as various organics and ethylenediaminetetraacetic acid (EDTA) or due to pH fluctuations. In addition, the stability of metalloenzymes can be affected by buffer complexations with metals. The free metal ion concentration in environmental systems is the best predictor of both bioaccumulation and toxicity of cationic metals [17,18]. Therefore, the bioavailability and toxicity of metals for microbes are dependent on (i) the ionic strength of a medium, (ii) the presence of organic matter, (iii) pH, (iv) redox potential, and (v) valence state. All these factors may favor the formation of different metal species with high or low bioavailability and toxicity [19].

Although MOB and AOM are substrate-specific, preferring CH_4_ and NH_3_, respectively, they can interact with one another in various ways [20]. Such interactions include the influence of ammonium on methane oxidation and MOB growth [21]. Moreover, MOB and AOB also share many physiological, structural, and ecological characteristics, including reliance on monooxygenase reactions catalyzed by the copper-containing membrane-bound monooxygenase superfamily (Figure 1 and see details below) [22], intracellular membrane systems, sensitivity to the same inhibitors, possession of hydroxylamine oxidoreductase systems, and the ability to grow in oxic environments [23]. Methane monooxygenase (MMO) in methanotrophs and ammonia monooxygenase (AMO) in ammonia oxidizers have evolved to be functionally identical, and they are capable of oxidizing both methane and ammonia [24]. Furthermore, interactions between methanotrophs and AOM and the effects of carbon and nitrogen cycles have rarely been studied in complex natural ecosystems because they are the critical member of global carbon and nitrogen cycles, respectively [23].

Copper is one of the important trace metals involved in various fundamental and specialized physiological processes, including electron transfer, oxygen transport, superoxide detoxification, denitrification, and ammonia and methane oxidation. Similar to other critical trace metals such as iron, copper is found in small amounts in the ocean, and is heavily complexed by organic ligands, which reduce the inorganic dissolved free metal [25]. It is suggested that Fe promotes the growth of various marine N-cycling microorganisms in a substantial section of oceans [26]. However, it is yet to be determined if metal availability influences the biological niche separation of AOM and MOB in the environmental systems. Therefore, it is crucial to know how AOM and MOB are affected by the fluctuations in metal concentrations, especially Cu and Fe, which will be covered in this review paper.

Collectively, since they are necessary for the enzymes involved in ammonia and methane oxidation, Cu and Fe are critically important for the growth and activity of microbes. However, high metal concentrations above the cell capacity could be toxic for microbes. In this review, we focused on aerobic ammonia and methane oxidation pathways mediated by metals and the impact of various metal concentrations. Especially Cu and Fe, on ammonia and methane oxidizers in in vitro and in situ systems, and different metal uptake strategies.

## 2. Ammonia and Methane Oxidation Pathways Mediated by Metal Compounds

Diverse ammonia- and methane-oxidizing microorganisms interact with each other in various ecosystems by coupling the oxidation reaction. They tend to utilize and sometimes compete with two different compounds, ammonia and methane, through copper-containing monooxygenase, which ultimately links the global carbon and nitrogen cycles.

### 2.1. Copper and Iron in the Ammonia-Oxidizing Pathway

Ammonia is oxidized to hydroxylamine (NH_2_OH) by ammonia monooxygenase (AMO) in all AOM, and then NH_2_OH is further oxidized to nitric oxide (NO) by hydroxylamine oxidoreductase (HAO) or cytochrome P460, which has been reported to be involved in the bacterial ammonia-oxidizing process (Figure 1) [27]. However, the exact mechanism of NO oxidation is still not identified, and some candidate proteins, such as a red copper protein oxidoreductase (NOO) in AOB and comammox, or reverse nitrite reductase (NIR) in all AOM, are assumed to be involved in this process. Comammox further oxidizes nitrite to nitrate (NO_3_^−^) by nitrate oxidoreductase (NXR; molybdenum containing enzyme) such as other nitrite-oxidizing bacterial (NOB) strains. However, ammonia oxidation mechanisms in AOA from NH_2_OH to NO_2_^−^ are still mysterious, and several unknown enzymes are supposed to be involved in this (Figure 1). The comammox strain *N. inopinata* carries *hao* gene encoding octa-heme cytochrome *c* protein which resembles the HAO protein of AOB. This *hao* gene shares a genomic locus with three other genes, including *haoB* gene coding putative membrane protein and two genes for tetra-heme *c*-type cytochromes, which resemble cytochrome *c*_554_ (CycA) and cytochrome *c*_m552_ (CycB) of AOB. Moreover, HAO, CycA, and CycB form the hydroxylamine ubiquinone reduction module (HURM) in AOB, which transfers electrons from hydroxylamine to the quinone pool.

Abundant vital proteins in AOM, notably the AMO, which accomplishes the energy-gaining step in ammonia oxidation, require copper as a cofactor [28,29]. Plastocyanins are copper-containing proteins that mediate electron transfer, encoded in high numbers by AOA. In addition, a high number of multi-copper oxidases and blue copper-proteins resemble genes detected in AOA genomes [28,29,30,31,32]. While a much higher number of Fe-based enzymes and fewer Cu-containing proteins are used in the ammonia-oxidizing system of the AOB mediated the electron transport system, which comprises heme-containing proteins such as HAO and CycB [29]. HAO and CycB have no homologs in AOA, and it is still unclear how these enzymes were replaced. However, Cu-containing plastocyanin is thought to operate as an electron carrier in place of CycB [33]. As a result, AOA should rely on Cu-containing enzymes more than their bacterial counterparts, necessitating complex regulatory mechanisms to respond to rapidly changing local Cu concentrations [34]. On the other hand, AOB uses more Fe-based enzymes and fewer Cu-proteins [35]. Therefore, AOB may have a lower Cu requirement.

As described above, the bioavailable free Cu^2+^ in the natural environment is substantially less than total copper because the Cu^2+^ speciation is dominated by complexation by dissolved organic carbon [36,37,38]. Therefore, the bioavailability of the free Cu^2+^ is highly dependent on complexation with organic matter. It has been identified that reducing copper bioavailability by metal-complexing organic compounds is a significant cause of AOA growth inhibition. AOA is considered much more sensitive than AOB to increased and decreased copper concentrations, and copper is a more critical trace element for AOA for ammonia oxidation and growth systems [34,39]. Therefore, it leads to the disproportion of AOA and AOB abundance in municipal activated sludge because of the different sensitivity of AOA and AOB to organic compounds [39]. Iron (Fe) is a vital element for AOB growth as it is related to energy generation and the functioning of enzymes in the electron transport chain, such as cytochrome *c*, which contains heme-*c* groups and needs to chelate with ferrous iron for active regions formation [40,41]. However, metals can also cause bio-accumulative toxicity because of their non-biodegradability, and accumulation in AOB leads to damage from reactive oxygen species (ROS) compounds; thus, too high concentration of Fe will also cause inhibition of ammonia oxidation (Figure 2). It has recently been found that a marine AOA strain *N. maritimus* has a lower Fe′ uptake rate (free Fe′ half-saturation constant; *K*_in_) in comparison to a marine AOB strain, a *Nitrosococcus oceani* strain C-107, and marine microorganisms [42]. Furthermore, the number of hypothetical Fe-binding sites in genome-predicted proteomes was significantly greater in marine AOB compared to AOA, and it is accorded that utilization of cytochromes burdens AOB with a significant additional Fe demand [42]. Contrariwise, a more significantly enriched Cu-binding protein in AOA than AOB is that AOA has an unusually high number of Cu proteins, including plastocyanins and multi-copper oxidases with a putative role in electron transport, as described above.

There are no studies that directly examined the role of the trace metal copper (Cu) and iron (Fe) in comammox bacteria growth. The iron requirement for a comammox strain may be the same with AOB because the ammonia oxidation and electron transfer machinery system are significantly similar. Furthermore, unlike canonical *Nitrospira* nitrite-oxidizing bacteria (NOB), the copper resistance proteins CopCD in comammox genome are highly similar to the homologs in beta-proteobacterial AOB and are located close to the AMO. CopAB is also detected in comammox *Nitrospira,* but it is not common for AOB. These proteins confer higher Cu^2+^ tolerance or increased Cu^2+^ uptake and [43], therefore, comammox *Nitrospira* may have higher requirements for copper than canonical *Nitrospira* NOB. Comammox may benefit from surviving in iron-limited environments because comammox genomes contain genes of the cytochrome *c* biogenesis system I, which has hemes with higher iron affinity [43]. In contrast, canonical *Nitrospira* NOB uses the cytochrome *c* biogenesis system II, which requires less energy for cytochrome synthesis [44]. Fe also participates as Fe-S clusters, utilized for electron transport and catalysis [45]. As bacterial nitrifiers have a high iron requirement for their Fe-S cluster- and heme-containing enzymes [43], this could provide an essential competitive advantage for comammox *Nitrospira* than canonical NOB *Nitrospira* in iron-limited environments [46]. Interestingly, the performance of comammox *Nitrospira* remained unaffected in a copper-limited wastewater system, and it even requires less copper than AOB [43]. Nonetheless, comammox *Nitrospira* may still need copper from natural environmental concentrations [47].

Other known ammonia oxidation processes include anammox [48], heterotrophic, and fungal nitrification process. Anaerobic anammox oxidation reaction include conversion of NO_2_^−^ and NH_4_^+^ to N_2_ and 2H_2_O, which is carried out in the anammoxosome, a membrane-bound compartment inside the cytoplasm. Despite the fact that anammox reaction can occur in two different ways, both use the same intermediate chemical, NH_2_OH, which is reduced from NO_2_^−^, and then interact with ammonia and are subsequently converted to hydrazine. Hydrazine is finally oxidized to N_2_ in the periplasm [49,50]. Anammox bacteria are known to have high ammonium, and nitrite affinity which make them suitable for substrate conversion at low concentrations [3]. Moreover, anammox is an essential process in the WWTP to remove nitrogen compounds. Anammox bacteria anaerobically convert ammonium and nitrite directly to N_2_, in the absence of aeration and other electron donors, which distinguish this process from heterotrophic denitrification. Anammox and aerobic AOM commonly co-exist in WWTPs systems, as nitrite is produced by the canonical ammonia oxidation, which can be utilized by anammox. However, due to their different oxygen and nitrogen sensitivity and activity, they need mutual cooperation for stable and efficient performance [51,52,53].

Heterotrophic nitrification is another method of oxidizing ammonia carried by heterotrophic nitrifiers, but the physiology and biochemistry of the relevant pathways is poorly known. Some heterotrophic ammonia oxidizers have an AMO system, but there are also non-AMO containing types of heterotrophic ammonia oxidizers [4]. Furthermore, heterotrophic nitrifiers use organic compounds as their energy source, in contrast to chemolithotrophic nitrifiers. One more ammonia oxidation process is fungal nitrification which is not identified clearly yet. AMO has not been identified in the fungal oxidation system [4].

### 2.2. Copper and Iron in the Methane-Oxidizing Pathways

Methane is the sole source of carbon and energy in methanotrophs. Methane-oxidizing bacteria and anaerobic methane-oxidizing archaea spontaneously oxidize methane [54]. There are three main groups of aerobic methanotrophs: type I, gamma-proteobacteria, which are found in the families Methylococcaceae and Methylothermaceae encode particulate methane monooxygenase (pMMO); type II, alpha-proteobacteria, which are found in the families Methylocystaceae and Beijerinckiaceae encode both pMMO and soluble methane monooxygenase (sMMO); and type III, Verrucomicrobia, which are members of the family Methylacidiphilaceae encode pMMO. Finally, *Methylomirabilis oxyfera*, a member of the NC10 phylum and the only known anaerobic denitrifying methanotrophic bacteria [55], utilizes an intra-aerobic pathway through the reduction in nitrite via a unique oxygen-producing pathway [56]. In order to use methane for energy and cell assimilation, methanotroph enzymes oxidize it to carbon dioxide through a series or particular pathways of linked reactions (Figure 1) [57].

The first step of methane oxidation on all known aerobic methanotrophs is started by methane monooxygenase (MMO; pMMO or sMMO), a crucial enzyme to oxidize methane (CH_4_) into methanol (CH_3_OH). The next step is the oxidation of methanol to formaldehyde (CH_2_O), which can either be converted to biomass or further oxidized to formate and carbon dioxide (Figure 1) [58]. pMMO is found in almost all methanotrophs and is located in intracytoplasmic membranes (ICMs). The sMMO and pMMO are mixed function oxidases that incorporate one oxygen atom into methanol and the other into H_2_O. These enzymes require two electrons and two protons to function [58]. It has been reported that MOB cultures expressing pMMO typically show a higher affinity for methane than cells that express sMMO [59]. In a classical aerobic MOB system, a cytochrome *bc*_1_ complex contains three heme groups and one Fe_2_S_2_ cluster, which is the electron donor to pMMO [60], based on similarities of pMMO to AMO in the AOB system. Therefore, even though pMMO only has two mono-copper sites [61], Fe is also required for pMMO activity [62]. However, the type 2 alpha-proteobacterial pMMO acquires electrons from the ubiquinol (Q_8_H_2_) through NADH oxidation [45]. In contrast, in gamma-proteobacterial methanotrophs, the oxidation of methanol to formaldehyde is also regarded as pMMO activity. Furthermore, it has been demonstrated that cells using pMMO for growth display higher growth yield, suggesting that the pMMO is the more efficient system for methane oxidation [63]. Interestingly, several MOB strains create sMMO that contains Fe under copper-limiting circumstances, while when copper is abundant, MOB expresses pMMO [64]. In the case of sMMO, it contains a di-Fe active site cluster in the hydroxylase component and an additional Fe_2_S_2_ cluster in a reductase subunit [64], and electrons would be donated by NAHD [65]. After oxidation of CH_4_ to CH_3_OH, methanol dehydrogenase (MDH) oxidizes CH_3_OH to formaldehyde (CH_2_O) using cytochrome *c*_L_ and *c*_H_ proteins as electron acceptors, which is oxidized by cytochrome *aa*_3_, that is heme-copper protein which contains two heme and copper [66]. MDH (MxaF-MDH) contains Ca^2+^ as a catalytic cofactor but recently discovered XoxF-MDH is present in yeast, fungi, and non-methylotrophic bacteria, depending on rare earth elements (REEs). In subsequent steps, oxidation from CH_2_O to CO_2_ is complex, and multiple pathways depend separately on the different methanotroph types [59]. During these steps, most enzymes primarily rely on Fe, with minor requirements of Cu, Ca, Mo, and Zn ions. In particular, the final step in the conversion of CHOOH to CO_2_, which contains the Fe-S cluster, is catalyzed by the membrane-associated enzyme formate dehydrogenase (FDH). The step from CH_2_O to CHOOH in *M. capsulatus* Bath is mediated by either a membrane-bound cytochrome-linked formaldehyde dehydrogenase (DL-FADH) which is mediated by high Cu (>1 µmol of Cu per mg of cell protein) or a soluble NAD(P)^+^-linked enzyme (N-FADH), which is mediated by low Cu [67]. As an electron acceptor, cytochrome *bc*1 (with Fe) is required for FADH. Therefore, Cu and Fe are the critical metal compounds that activate methane oxidation.

### 2.3. Ammonia Oxidation by MOB and Methane Oxidation by AOM

Many connections exist between methanotrophic and nitrifying microorganisms. Methane monooxygenase (MMO) is tightly linked to AMO, although inefficiently, it can oxidize ammonia to hydroxylamine [68]. Since ammonia (NH_4_^+^) and methane (CH_4_) are structurally similar, AMO in nitrifiers may also oxidize methane, but the oxidation activity is less efficient than MMO in methanotrophs [1,69]. CH_4_ is a nonpolar molecule as it has a symmetric tetrahedral geometrical shape with four identical C-H bonds. Ammonium ion (NH_4_^+^) is a positively charged ion (cation) and is also nonpolar in nature based on its tetrahedral structure. On the other hand, ammonia is a polar molecule, its polarity is induced by the electronegativity difference between N (3.04) and H (2.2) atoms. Interestingly, it has not been clearly identified yet which form of ammonia (NH_4_^+^ vs. NH_3_) is the actual substrate for AOM, but recently it has been proposed by a kinetic experiment that ammonia (NH_3_) rather than ammonium ion (NH_4_^+^) is the actual substrate for all AOM [70]. Nevertheless, the pMMO and AMO enzymes are located in the periplasmic membrane; therefore, the methane and ammonia could be active without passive transportation. However, sMMO is located in the cytoplasmic space, so methane should be transported into the cytoplasm for the sMMO (see above).

In the initial phase of aerobic oxidation, Cu-containing monooxygenase (MMO) enzymes combine a single oxygen atom from O_2_ into CH_4_ or NH_3_ (see above) to produce methanol from CH_4_ and hydroxylamine from NH_3_ (Figure 1) [23]. Both of these microbes can co-oxidize various substrates and are blocked by the same chemicals [23]. Methanotrophs have been demonstrated to engage in soil nitrification in nutrient-limited circumstances [71]; however, aerobic MOB cannot grow by either NH_3_ or NH_2_OH oxidation. Therefore, they must rely on methane (or methanol) oxidation for ammonia oxidation [23]. Although NH_2_OH is an intermediate of ammonia oxidation, it is highly toxic; therefore, both ammonia-oxidizers and MOB have to remove it quickly. In the bacterial ammonia oxidizer, electrons produced from hydroxylamine oxidation either by a two *c*-type cytochrome and hydroxylamine oxidoreductase (HAO) or cytochrome P460 to the quinone pool for energy generation and cell growth (see above and Figure 1) [27], but MOB lack this system. Therefore, the hydroxylamine is oxidized by detoxifying activity without generating electrons [72]. Many methanotrophs encode a HAO-like protein, and these HAO-like proteins carry out hydroxylamine oxidation in methanotrophs, but biochemical evidence has not yet been identified [73]. Up-regulation of HAO transcription in response to ammonia has been shown for several methanotrophic bacteria that oxidize ammonia faster than those not encoding HAO genes [71,74,75]. During the step of hydroxylamine oxidation by MOB, nitric oxide (NO) would be produced as a key intermediate, similar to AOM (Figure 1). Recently, it has been reported the catalytic properties of the HAO in thermophilic *Verrucomicrobia* methanotroph (MOB type III), and the HAO has a crucial function of rapid oxidation of NH_2_OH to NO and help methanotrophs thrive in environments where methane and ammonia coexist [76].

The other way around, methane oxidation by AOM is also reported in some AOB strains, *Nitrosomonas* and *Nitrosococcus* [77,78,79]. For example, in the ^14^CH_4_ trace experiment, the *Nitrosomonas* and *Nitrosococcus* AOB strains oxidized a significant amount of CH_4_ and incorporated ^13^C into cellular components. The ability of AOM to use both ammonia and methane could be a profitable strategy when one or the other is limited and cannot be used as an energy source. Since methane is nearly always present in natural environmental systems, the survival advantage of using either substrate is obvious. However, the precise methane oxidation mechanism of AOB has not been identified. Furthermore, the possibility of methane oxidation by other members of AOA, and comammox, has not been demonstrated yet.

### 2.4. Environmental Pollutants with Ammonia and Methane Oxidizer

In AOB-dominant conditions, organic micropollutants are bio-transformed to nitrogen-containing compounds, such as hydroxylamine, nitrite, and nitric oxide. Furthermore, In autotrophic reactors, the contaminant was quickly degraded with high efficiencies than in non-AOB-dominant microbial communities [80]. Micropollutants might activate AMO by positing into the hydrophobic pocket and reacting with the oxygen-activating site, allowing micropollutants to be oxidized in the presence of ammonia [81]. It has been reported that an AOA strain *Nitrososphaera gargensis*, could biotransform numerous micropollutants, including tertiary amines, mianserin, and ranitidine, during ammonia oxidation [82]. Furthermore, the heavy metal transport genes found in AOA genomes suggest that they are adaptable to heavy metal pollution [83].

Methanotrophs have also been reported to decompose various heavy metals and contaminants, including hydrocarbons, and halogenated organic compounds, in the presence of methane monooxygenase (MMO) [84,85,86]. Furthermore, it has been demonstrated that pMMO protein could have the activity to oxidize alkanes and alkenes, while soluble sMMO can oxidize aromatics, aliphatic compounds, and a wide range of organic compounds [87,88]. Therefore, it is worth revisiting the physiological pathway of ammonia and methane oxidation to verify which enzymatic function could connect to the pollutant degradation or biotransformation containing metal compounds.

## 3. Ammonia and Methane Oxidizer Reactions with Metal in Elevated Metal Concentrations

Metals are essential ecosystem components, and their biological availability is primarily determined by geological and biological processes [89]. As described above, bioavailable metals serve metabolic purposes as enzyme ingredients or structural functions such as maintaining the cell envelope. Accordingly, it would be challenging for microbes to uptake the bioavailable metal from the ecosystem in limited metal concentrations. Harmful metal effects are supplanting the enzymes-associated metals with similar structures or enzyme damage and inactivation by high concentrations (Figure 2) [90]. Therefore, different concentrations of metals may be key control factors for the microbial community and abundance, and as a result, the environmental system may exhibit microbial niche differentiation.

### 3.1. High Concentrations of Metals

As described above, the bioavailable free metal concentration is tightly affected by various factors. Total metal concentration is increased by the input of the environmental pollution that has become a problem, and it is a significant concern due to the adverse effects it is causing worldwide. Inorganic pollutants are being discarded in both the terrestrial and the aquatic environment due to the astounding increase in the use of heavy metals for anthropogenic activity, which is the primary cause of pollution. Heavy metal pollution is also caused by agricultural activity, such as pesticides, insecticides, fertilizers, etc. Since metals are non-biodegradable and remain persistent in the environment for a long time, they cannot be cleared. Therefore, heavy metals in the environmental system remain present for long periods until they are eluted to other compartments, and then the accumulated metal becomes more toxic to microbes. Excess high concentration of metals is toxic in (i) the displacement of essential metals from their regular binding sites of metalloenzyme, (ii) disruption of nucleic acid structure leads to inhibition of enzymatic functioning, (iii) genotoxicity, (iv) oxidation stress (production of ROS), and (v) inhibition of signaling pathways [91] (Figure 2). In other cases, organic ligands and metal complex formation may effectively reduce the metal concentration below a toxic threshold. Adding metal chelators may alleviate the harmful effects of metal because toxicity is related to the free metal concentration, which is critical for metal toxicity. For instance, if copper were not complexed in the environment, copper-sensitive microbes, many members of autotrophs, would be exposed to toxicity due to high copper concentrations. On the other way around, organic pollution could decrease metal bioavailability by forming a complex between metal and organic ligands, which inhibits the growth and activity of oligotrophs such as AOA in the various heterotrophic environments [39]. Therefore, the equilibrium between the predominance of chelators and metals determines the bioavailability of the metal concentration necessary for the growth of different microorganisms.

A high level of heavy metal compounds had a significant negative impact on both environmental nitrification activity and diversities of AOM. Therefore, several previous studies demonstrated the complexation of copper and other metals with organic matter in activated sludge that focuses on metal toxicity for microbially mediated processes [92]. It has revealed that increasing concentrations of metal (320X of original trace metal solution), including Cu and Fe, above the capacity of the cell, clearly decreased ammonia-oxidizing activity of both AOA and AOB strains, even though there were chelating agents in the media [39]. Furthermore, *Nitrosomonas* AOB was much more resistant to copper amendment than other tested AOA strains [39]. However, short-term laboratory microcosm experiments found that AOA were consistently more abundant than AOB, and there was no significant AOA community shift after treatment with different concentrations of Cu, either with or without Cu [93]. In another study, the abundance of AOA was lower than recovered AOB abundance after 2-year Zn exposure, and AOB played a significant role in the restoration of nitrification rather than AOA [94]. Therefore, these results indicated that the deleterious effect of metal compounds on the abundance, diversity, activity, and composition of ammonia oxidizers in the natural environment was closely related to the element types, contents, metal exposure times, ecophysiological conditions, etc.

Methanotrophic niches containing pMMO are found in soils, sediments, lakes, and oceans at a high concentration of copper. In contrast, excessive copper concentrations inhibit pMMO activity, resulting in the generation of hydrogen peroxide, which reversibly inhibits pMMO [95,96]. In the culture of *Methylococcus capsulatus* Bath, pMMO facilitates CH_4_ oxidation in the cytoplasm when Cu/cell ratios are high (increased from 0 to 55 µM of Cu), while sMMO is activated in the periplasmic membrane space when Cu/cell ratios are low (decreased to 0 µM of Cu) because of different expression patterns (see above). Any other metal ion except copper does not control this switch; however, precisely how copper regulates pMMO expression is still unknown. This copper switch is regulated by methanobactin (Mb), a chalkophore (such as a siderophore; see below). The chelation of copper forms the Cu-Mb complex, which is a copper uptaking strategy of MOB in copper-limited conditions [97]. On the other hand, Cu-chelating Mb also played a critical role in maintaining Cu homeostasis and protection against the potentially toxic effects of a high Cu concentration (*M. trichosporium* OB3b on 10 µM copper) (see Figure 2) [98]. In fact, the primary function of Mb is to bind and accumulate high-affinity Cu(I) atoms in the growth media when copper concentration is limited.

### 3.2. Low Concentrations of Metals

Various organic chemicals inhibit the growth of AOA and AOB in sterile-filtered wastewater. AOA was more susceptible to organic chemical inhibition than the tested AOB representative [99]. As described above, the copper complexation with organic compounds significantly inhibited AOA growth, implying that differences in copper requirements and acquisition methods between AOA and AOB likely explain their different sensitivity to organic compounds [39]. Furthermore, copper supplements significantly diminish AOA inhibition by organic compounds, allowing an AOA strain to flourish in municipal nitrifying activated sludge [39]. The effect of ammonia removal was also enhanced by copper addition in a copper-limited full-scale groundwater treatment bioreactor [47]. Therefore, a metal-restricted environment not only loses biological function but also reduces ammonia oxidation activity in various environmental systems.

The bioavailability limitation of divalent metals by complexation with organic matter could be explained by the stability constants (*K*) for various divalent metal-organic complexes (see above). Free copper (Cu^2+^), the dominant form of dissolved copper in oxic water, is situated at the top of the Irving-Williams Series, leading to its highest affinity for most environmental organic ligands [100]. Cysteine and histidine being highly inhibitory to AOA, have the highest *K* values with Cu^2+^. Organic matters composed of fresh and partially decomposed (e.g., amino acids, sugars, and peptides) and well-decomposed (e.g., humic acids) organic compounds can form tight complexes with Cu^2+^ [101,102,103]. Thus, the bioavailable form of copper compound (Cu^2+^) is limited in various environments because most Cu^2+^ (98–99%) have been found in complexes with organic matter [104,105,106]. Then the different enzyme affinities, together with the different quantities of bioavailable metals, create niche differentiation between various ammonia oxidizer types. Low copper bioavailability can only affect the growth and activity of microbes in the absence of siderophore and chalkophore systems. It is a significant constraint to ammonia and methane oxidation in urban wastewater treatment plants (WWTPs) [107], as well as in many other organic-rich ecosystems, and copper bioavailability is a crucial element in the niche difference between different groups of ammonia oxidizers [39]. Furthermore, when Cu^2+^ bioavailability is lower than the Cu^2+^-limiting threshold of AOA [106], microbial Cu^2+^ dependency can impact the nitrogen cycle [33]. In the WWTPs, metal compounds and other factors such as dissolved oxygen (DO) and sulfate lead to activity and community change of AOB, sulfate-reducing bacteria, anammox, and anaerobic heterotrophic bacteria. Therefore, these multiple factors should be considered together with controlling metal compounds to enhance the efficiency of wastewater treatment [108,109].

Higher pH increases metal binding to organic and inorganic soil particles, dissolved organic molecules, or dissolved minerals [110]. The reduction in copper and iron in these soil environments can affect the abundance and activity of AOM and MOB. In acidic environments, where copper is more bioavailable than iron, the physiological properties of AOA may provide a competitive advantage over AOB [111]. According to the findings of the soil community investigation, reduced pH to AOA strains are considerably more crucial to contributing to environmental nitrogen cycles [112,113]. *N. viennensis*, a soil AOA strain, had reduced ammonia oxidation and growth when Cu was limited, which was accompanied by the downregulation of the genes involved in metabolism for electron transport, carbon fixation, nucleotide, and lipid biosynthesis [34].

When copper is limited, MOB releases the Mb, which is a class of copper-binding metallophores known as chalkophores mediating microbial copper homeostasis, functioning similar to siderophores which are involved in iron homeostasis (see below in details) [114]. Under metal-limited conditions, various microbes tend to uptake the metal compounds needed for the active site of the key functional protein, such as MMO and FDH in MOB [95,115]. As previously mentioned, variable copper concentrations could control the differential expression of pMMO and sMMO.

## 4. Chalkophores and Siderophore

Some microorganisms produce siderophores, which are high-affinity Fe binding ligands that increase Fe bioavailability. The siderophore-complexed Fe is subsequently transported into the cytoplasmic space of the cell [114]. Various types of siderophore are transported into cytoplasmic space by siderophore-specific membrane channels or by reduction in Fe-chelates, and it provides a competitive advantage in iron-poor niches of the environments [26]. Ammonia oxidizers, especially bacterial ammonia oxidizers, AOB and comammox, are known for their significant Fe requirement because many cytochromes and heme-containing enzymes are necessary for the energy metabolism (see above). However, the AOB strain *Nitrosomonas europaea* lacks a genome-based system for producing its own siderophores. Instead, it encodes a large number of genes for iron acquisition enabling efficient scavenging and uptake of various forms of iron in iron-limited environments [35]. Interestingly, oligotrophic AOB strain *Nitrosomonas eutropha* and marine AOB strain *Nitrosomonas oceani* have the aerobactin biosynthesis pathway in the genome (Table 1) [116,117]. In vitro investigation on siderophore production by any AOM have not yet been conducted. In contrast to *N. europaea*, which possesses multiple classes of σ factor/anti-σ factor/TonB-dependent outer membrane (OM) siderophore transporter as well as regulatory genes [118], *N. eutropha* has only a single ferric uptake regulator gene. Thus, *N. eutropha* likely relies on different mechanisms compare to *N. europaea* to regulate iron uptake. Interestingly, the Fe^3+^-siderophore transporter of *N. europaea* responded to iron limitation, which is generally repressed under iron-replete conditions. *N. eutropha* contains conserved genes encoding the energy-transducing TonB-protein complex but lacks an ABC transporter for Fe^3+^/siderophores. Therefore, *N. europaea* may utilize siderophores produced by other organisms in its environmental consortium. The ability of the strain to survive in an iron-limited environment without costly secretion of reduced carbon and siderophore could be advantageous. For instance, NOB *Nitrobacter* and *Nitrospira* can produce siderophores, and that *N. europaea* presumably uses these siderophores, hence the AOB *N. europaea* and NOB strains commonly live together as a consortium in ecosystems [119]. However, smaller genetic inventories for iron transport and the presence of putative siderophore biosynthesis genes in other AOB such as *N. eutropha* and *N. oceanii* indicate that they possess different iron acquisition strategies. It has been recently reported that marine AOA strain *N. maritimus* can utilize strongly chelated Fe to the siderophore desferrioxamine B mesylate (DFB), using an extracellular reductive uptake strategy, even though the AOA strain lacks an endogenous siderophores biosynthesis and transporter system (Table 1) [26].

As described, chalkophores are high affinity copper-complexing agents that are secreted by particular bacteria and form solid Cu complexes [115,123]. Most methanotrophs produce Mb to facilitate copper uptake, maintain Cu homeostasis, and protect against the toxic effects of high Cu concentration (*M. trichosporium* OB3b on 10 µM Cu) [98]. Mb chelate and scavenge Cu^2+^ or Cu^+^ from the environment and shuttle them into the bacterial cells. In MOB, metals are transported into the cell by passive diffusion or active transport through TonB-dependent metal transport or other metal transport proteins; in the case of the Mb-Cu complex are transported by the active transporter systems [124]. Therefore, methanotrophic activity in nature may be controlled and influenced significantly by the capacity of Mb to acquire Cu even from insoluble surroundings such as minerals [125]. As mentioned above, Mb maintains Cu homeostasis and protects the cells from metal toxicity caused by the high concentration of Cu as well as other metals. For example, Mb has the ability to bind Au(III), Hg(II), although its binding constant is 15 parts lower in magnitude than copper [114]. However, it has not been verified if interactions between Mb and these metal compounds are relevant to the activity of methanotrophs. Mb is also related to binding other metals such as Fe(III), Ni(II), Zn(II), Au(III), Ag(I), Pb(II), Mn(II), Cd(II), and Co(II), but the binding constants of all these metals examined were less than Cu(II) except Ag(I) and Au(III) [121]. Conversely, some groups of siderophore could bind other metals rather than Fe with relatively high affinity, however, the copper-binding capacity is not regarded as copper uptake. Therefore, it might be a biologically relevant function of several siderophores [114].

## 5. Conclusions and Future Study

High concentrations of metals in the environment are introduced from metal-containing inorganic pollutions produced by anthropogenic activity, which has a substantial impact on microbial communities and alters their activity. Understanding carbon and nitrogen cycling at the molecular level may also allow insight into how microorganisms are adapting and benefiting from various metal concentrations at the ecosystem level. This is an important step toward mitigating the further release of greenhouse gases containing carbon and nitrogen into the atmosphere. Aerobic methane and ammonia oxidizers (MOB and AOM) are critical members of the carbon and nitrogen cycle in aerobic environments. Their metabolic capacity is significantly affected by metal compounds. Our review concluded that copper and iron metals are essential for various fundamental and specialized physiological processes in MOB and AOM. They are vital in the functioning of enzymes involved in the oxidation processes, which facilitate the growth and activity of microbes in various environments. On the other hand, high metal concentrations are considered toxic for microbial communities. However, further studies are still needed to address the unsolved question in our knowledge of trace metal physiology. For example, (i) comprehending the uptake system of metal or complex of metal-ligands by AOM, (ii) the ammonia oxidation activity affected by methanobactin on AOM growing culture in limited or excess metal conditions, (iii) co-metabolism of methane oxidation by AOA and comammcox with different metal concentrations, and (iv) relationships between metal bioavailability and fluxes of greenhouse gases produced by MOB and AOM. This review study could provide essential information for future works, not only to answer these fundamental questions regarding the ecology and physiology of MOB and AOM in diverse ecosystems but also for comprehending and managing C and N cycles in nature.

## Figures and Tables

**Figure 1 life-12-01806-f001:**
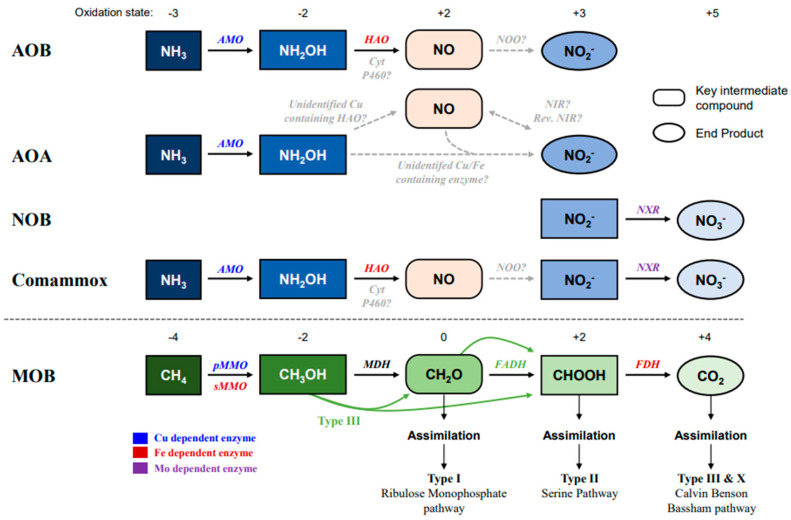
Pathway of ammonia and methane oxidation by ammonia oxidizers (AMO: AOA, AOB, and Comammox), NOB, and methane oxidizer (MOB). The blue color indicates the oxidation of ammonia (NH_3_), and the green color indicates methane oxidation. Black straight arrows indicate identified reactions, and grey dotted arrows show unknown enzyme activity or unidentified pathways. Enzymes at the above arrows with different colors indicate different metal dependencies. The oxidation states are at the top of the pathways. Abbreviations: AMO, Ammonia monooxygenase; HAO, Hydroxylamine oxidoreductase; NXR, Nitrite oxidoreductase; pMMO, Particulate methane monooxygenase; sMMO, Soluble methane monooxygenase; MDH, methanol dehydrogenase; FADH, formaldehyde dehydrogenase; FDH, format dehydrogenase.

**Figure 2 life-12-01806-f002:**
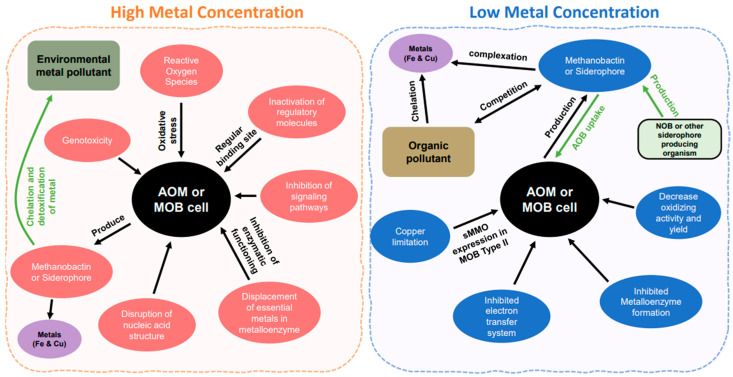
Metal effects between the high and low concentrations. The arrows indicate the direction to the cell or from the cell. In high concentration, pink color is showing the effect of high concentration of metal within the cell. Green and purple color showing metals outside the cell or in the environment. In low concentration, blue color is indicating the disruption within the cell. Brown color represent metals in the environment and green color shows microorganisms other than ammonia and methane oxidizers.

**Table 1 life-12-01806-t001:** Siderophore and chalkophore of ammonia and methane oxidizer.

Scheme	Siderophore Biosynthesis	Siderophore Transporter	Chalkophore Biosynthesis	Chalkophore Transporter	Note
*Nitrosococcus oceani* ATCC19707	+	+	N.D.	N.D.	*Nitrosococcus oceani* has hydroxamate-type siderophore aerobactin synthesis and aerobactin receptor genes, and the iron transporter contains an ABC-type Fe^3+^/cobalamin siderophore transport system [116].
*Nitrosomonas europaea* ATCC19718	−	+	−	−	*Nitrosomonas europaea* has many iron-related genes to obtain iron from the environment; one of them is siderophore. However, no evidence was found for siderophore production [118].
*Nitrosopumilus maritimus* SCM1 ^a^	−	−	−	−	Most ammonia-oxidizing archaea do not have siderophore and chalkophore-related functions, but only *N. maritimus* SCM1 has a siderophore-related gene.
*Nitrosocosmicus oleophilus* MY3	−	−	−	−	
*Nitrososphaera viennensis* EN76	−	−	−	−	
*Nitrospira inopinata*	± ^b^	+	−	−	*Nitrospira inopinata* has putative TonB-dependent receptors (NITINOP_0357, NITINOP_1937, NITINOP_2355).
*Methylosinus trichosporium* OB3b	− ^c^	±	+	+	After binding copper, methanobactin is reinternalized through a specific outer membrane TonB-dependent transporter [114,120,121].

+, Exist the gene; −, Not exist the gene; ±, Still in controversy; N.D., No data. ^a^
*N. maritimus* can utilize strong chelated Fe to the siderophore desferrioxamine B mesylate (DFB), using an extracellular reductive uptake strategy [26]. ^b^
*Nitrospira* (NOB) produces siderophores that has not been identified in *N.inopinata* [46]. ^c^
*M. trichosporium* OB3b can produce iron chelate compounds, but the siderophore biosynthetic pathway is still unidentified [122].

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
