# Peer review of "Ecological Aerobic Ammonia and Methane Oxidation Involved Key Metal Compounds, Fe and Cu"

_life, 2022, doi:10.3390/life12111806_

Round 1

Reviewer 1 Report

It is very comprehensive study about the metal effects on ammonia and methane oxidation. However, it would only be necessary to include concentration values of cupper and iron metals or concentration ranges throughout the review. The words "high" and "low" appear 46 and 24 times, respectively. I think it is interesting for the reader to be able to quantify and not only have a qualitative view.

There are many abbreviations used in the work, so a list of them would be desirable.

L90, [11,12] instead [11][12]

L79. Add a reference.

L140-141. The reference 24 was published in 2008, it should be not considered as “recently”

L254, L425; L426. Checks the superscripts in the elements.

Author Response

Comments from reviwer: #1 (Comments to the Author):

It is very comprehensive study about the metal effects on ammonia and methane oxidation. However, it would only be necessary to include concentration values of cupper and iron metals or concentration ranges throughout the review. The words "high" and "low" appear 46 and 24 times, respectively. I think it is interesting for the reader to be able to quantify and not only have a qualitative view.

Re) Thank you for the positive comments. We added specific ranges of the metal concentration for various strains in some parts of the revised MS in response to the reviewer feedback. However, as we described in the MS, the physiological mechanism of enzymes and the growth of the microbes is mainly affected by free metal compounds, not the total metal concentrations. Furthermore, several previous studies we cited in the MS do not have clear information about the complexability between the metal and other compounds; therefore, it is not easy to identify the concentrations of free metals. Nevertheless, we tried to add the concentration information as much as possible, so please take into account the updated MS.

There are many abbreviations used in the work, so a list of them would be desirable.

Re) Thank you for the comment. In response to the reviewer's suggestions, we have made the list of the abbreviations. However, it is not sure it fits with the format of the Life journal.

L90, [11,12] instead [11][12]

Re) Corrected.

L79. Add a reference.

Re) Added

L140-141. The reference 24 was published in 2008, it should be not considered as “recently”

Re) Revised.

L254, L425; L426. Checks the superscripts in the elements.

Re) Corrected.

Reviewer 2 Report

In this review manuscript, the authors summarised ammonia and methane oxidation pathways mediated by metals, the impact of various metal concentrations, especially Cu and Fe on ammonia and methane oxidizers in in-vitro and in-situ systems, as well as the different metal uptake strategies. This review can provide valuable aspects for understanding the current research progresses on ammonia and methane oxidation. However, the core viewpoints are not clear enough. Necessary revisions are still required before its final publication.

1. The academic contribution of the review manuscript should be properly highlighted in the abstract, conclusion, and discussion.

2. The abstract needs to be rewritten.

3. The influence of metals on the oxidation of ammonia and methane and its practical significance need to be further discussed.

4. Ammonia and methane have different properties, such as dissolution and dissociation. Whether these differences are properly accounted for in this manuscript?

5. The interaction between microorganisms under different conditions (including the presence of metal ions) needs to be discussed with relevant research results, such as Bioresource Technology, 2022, 347, 126364; Bioresource Technology, 2019, 280, 173–182.

6. Different kinds of multi-metals may have different effects on the oxidation of ammonia and methane. Do they represent different types to be distinguished? Please check it.

7. Ammonia oxidation includes aerobic ammonia oxidation and anaerobic ammonia oxidation. Can summarise them apart?

8. More studies can be cited to support anaerobic ammonia oxidation, especially on the aspects of ammonia oxidation. Such as Bioresource Technology, 2022, 361, 127750; Bioresource Technology, 2022, 357, 127379; Journal of Environmental Sciences, 2023, 125, 160-170; Journal of Environmental Management, 2021, 296, 113271; Chemosphere, 2021, 278, 130436.

9. The types of ammonia-oxidizing microorganisms and their influence by metals need to be further summarized.

10. Suggestions on improving the format and grammar. Such as Line 101: “CH4 and NH3”; Lines 131-132 and 487-488: the line space?

Author Response

Comments from reviwer: #2 (Comments to the Author):

In this review manuscript, the authors summarised ammonia and methane oxidation pathways mediated by metals, the impact of various metal concentrations, especially Cu and Fe on ammonia and methane oxidizers in in-vitro and in-situ systems, as well as the different metal uptake strategies. This review can provide valuable aspects for understanding the current research progresses on ammonia and methane oxidation. However, the core viewpoints are not clear enough. Necessary revisions are still required before its final publication.

Re) We thank the reviewer for the valuable comments on the significance of our manuscript and for the constructive suggestions below.

  1. The academic contribution of the review manuscript should be properly highlighted in the abstract, conclusion, and discussion.
    Re) As reviewers pointed out, we have highlighted the academic contribution of the manuscript in the abstract and conclusion section. Please check the updated points.

  2. The abstract needs to be rewritten.
    Re) Thank you for the comment, we have revised the abstract to highlight the significance of the review manuscript.

  3. The influence of metals on the oxidation of ammonia and methane and its practical significance need to be further discussed.
    Re) We appreciate the comment. Copper and iron are essential because they are needed for key enzymes involved in the first step of ammonia and methane oxidation. This is well documented in the sections ‘Copper and iron in the ammonia-oxidizing pathway’ and ‘Copper and iron in the methane-oxidizing pathways’. Additionally, we summarized the effects of different metal compounds and various metal concentrations on the methane and ammonia oxidations. In the ecosystems, the metal concentration is elevated due to various causes, which significantly affect positively or negatively ammonia and methane oxidation, and it ultimately affects the global N and C cycle. And it is described in the manuscript in detail in the sections ‘High concentrations of metal’ and ‘low concentrations of metal’. We also changed the title and many parts of the manuscript to describe the main purpose of the review study more clearly.

  1. Ammonia and methane have different properties, such as dissolution and dissociation. Whether these differences are properly accounted for in this manuscript?
    Re) Thank you for this valuable comment. Since ammonia (NH4+) and methane (CH4) are structurally similar, AMO in nitrifiers may oxidize methane. In the case of MOB, methane monooxygenase (MMO) is tightly linked to AMO, although inefficiently, it can also oxidize ammonia to hydroxylamine. In the initial phase of aerobic oxidation, Cu-containing monooxygenase (MMO) enzymes combine a single oxygen atom from O2 into CH4 or NH3 to produce methanol from CH4 and hydroxylamine from NH3. These microbes can co-oxidize various substrates and are blocked by the same chemicals. CH4 is a nonpolar molecule as it has a symmetric tetrahedral geometrical shape with four identical C-H bonds. The electronegativity of carbon and hydrogen is 2.55 and 2.2, respectively, which causes the partial charges to be almost zero. The ammonium ion (NH4+) is positively charged (cation) and is also nonpolar in nature because of its tetrahedral structure. In the NH4+ molecular structure, all four hydrogen atoms are symmetrically bonded to the nitrogen atom and cancel out the polarity of the N-H bonds, resulting in a nonpolar NH4+ On the other hand, the electronegativity difference between N (3.04) and H (2.2) causes the polarity of the NH3 molecule; therefore, ammonia is a polar molecule. Interestingly it has not been clearly identified yet which form of ammonia is the actual substrate for AOM, but recently it has been proposed by a kinetic experiment that the ammonia (NH3) rather than ammonium ion (NH4+) is the actual substrate for all AOM (Jung et al., 2022). Nevertheless, pMMO and AMO enzymes are located in the periplasmic membrane; therefore, the methane and ammonia could be active without passive transportation. However, sMMO is located in the cytoplasmic space, so methane should be transported into the cytoplasm for the sMMO. We have addressed it in the updated manuscript (L320-331), so please check the points.

  1. The interaction between microorganisms under different conditions (including the presence of metal ions) needs to be discussed with relevant research results, such as Bioresource Technology, 2022, 347, 126364; Bioresource Technology, 2019, 280, 173–182.
    Re) Thank you for the kind suggestions. We have discussed the points with the reference papers in our revised manuscript (L486-490).

  2. Different kinds of multi-metals may have different effects on the oxidation of ammonia and methane. Do they represent different types to be distinguished? Please check it.
    Re) Thank you for your comment. Copper and iron are the key metal compounds in ammonia and methane oxidation, and it has been well addressed in the MS. Interactions between microbes and other trace metals such as zinc (Zn), cobalt (Co), molybdenum (Mo), selenium (Se), manganese (Mn) is also essential in the growth and activity of microbes. These metals contribute as key or co-factors in nitrite oxidation and CH3OH oxidation to CHOOH via CH2 We described these points in the manuscript (highlighted in blue) and Figure 1. However, as we changed the title and highlighted it in the manuscript, we would like to focus on the key metal compounds, copper and iron, effect on aerobic ammonia oxidation. Because knowing how AOM and MOB are affected by these metal compounds, is critical for understanding and controlling the global C and N cycles.

  3. Ammonia oxidation includes aerobic ammonia oxidation and anaerobic ammonia oxidation. Can summarise them apart?
    Re) We thank the valuable comment. According to the comment, we updated the manuscript by describing other ammonia oxidation processes (see L238-261). However, under other environmental conditions compared to the canonical aerobic AOM, completely different physiological pathways participate in the oxidation process. Therefore, in this review, we focused on canonical aerobic ammonia oxidation, carried out in aerobic conditions, and it could be directly compared to the aerobic methane oxidation pathway which is physiologically similar.

  1. More studies can be cited to support anaerobic ammonia oxidation, especially on the aspects of ammonia oxidation. Such as Bioresource Technology, 2022, 361, 127750; Bioresource Technology, 2022, 357, 127379; Journal of Environmental Sciences, 2023, 125, 160-170; Journal of Environmental Management, 2021, 296, 113271; Chemosphere, 2021, 278, 130436.
    Re) Thank you very much for the suggestions and the literature. We cited some of the literature that explains the nitritation-anammox process in the section ‘Copper and iron in the ammonia-oxidizing pathway’ (L246-253). But, as answered above, our main purpose of the study is not the anaerobic oxidation of ammonia and methane.

  2. The types of ammonia-oxidizing microorganisms and their influence by metals need to be further summarized.
    Re) The content of the influence of metals on the different type of AOM are described in the section ‘Copper and iron in the ammonia-oxidizing pathway’. As we explained in the manuscript, different physiological pathways for the ammonia oxidation of the different types of ammonia oxidizers are making other effects on the metal compounds. Furthermore, the sensitivity of oxidizing activity and affinity by the influence of metals are also different. For example, AOA growth was significantly inhibited by copper complexation with organic compounds, suggesting that differences in copper requirements and acquisition methods may account for their different sensitivity to organic compounds. On the other hand, Iron (Fe) is a much more vital element for AOB growth than AOA as it is related to energy generation and the functioning of enzymes in the electron transport chain, such as cytochrome c, which contains heme-c groups and needs to chelate with ferrous iron for active regions formation. Unfortunately, there are no studies that directly examined the role of the trace metal copper (Cu) and iron (Fe) in comammox bacteria growth. But we have described copper and iron requirement of the comammox compared to AOA and AOB systems. So please check and consider these points

  3. Suggestions on improving the format and grammar. Such as Line 101: “CH4 and NH3”; Lines 131-132 and 487-488: the line space?
    Re) Thank you for the intensive review. We have corrected it.

Reviewer 3 Report

The present study investigated the the metabolic diversity of  ammonia and methane oxidation related to metal compounds and the effect of various concentrations of metal compounds on MOB and AOM, which are evolutionally connected, enriched with different functional groups of bacteria.

Present review is short introduction to N and C cycle. Only one table has been added, some figures could be necessary to show distinct process differences. Nitrogen removal rates could be added as well.

 This paper presents the latest achievements on the nitrogen removal and the utilization of biodegradable organic matter to provide methane and ammonia for AOB, Comammox whereas anammox process contributing important nitrogen cycle driver in marines has not been outlined. Optimal condition for AOB and MOB process could be given to be reached. Processes (ammonia oxidation to nitrate and nitrite) during an anaerobic-anoxic operation in a system was not outlined achieved.

Few remarks can be made on MS quality:

1. “The oxidation of ammonia initiates nitrification by ammonia-oxidizing microorganisms (AOM), which is mediated by three distinct groups of aerobic autotrophic ammonia oxidizers: i) ammonia-oxidizing bacteria (AOB), ii) ammonia oxidizing-archaea (AOA), and iii) complete ammonia-oxidizing bacteria (Comammox) “ The process sequence to achieve TN after effluent treatment through AOB and anammox process seems complicated and is not  sufficiently outlined, whereas also in real application many things have been done lately in these process applications. Complicated control of metals and dissolved oxygen (for MOB) and the instability of ammonia feeding at right ratio in bioreactors treating high NH4+ -N wastewater make complications in practice need to be outlined. Elaborate

-Out as total studies metal toxicity to AOM, Comamox and denitrification is also extensively studied. Why deni and anammox was not studied thoroughly. There is bad wording in some sentences

2. Work showed a amount of ammonia to be recycled by the use of Cyt, results through Comammox, which efficiencies are needed to give – Give standard deviation of your resultant pollutants’ removals, mean, max and min in separate parts available in literature.

3. Needs descriptions about the exact metal parameters observed and given are to be toxic and why instabilities in current metals at their exact concentrations over how short time occurrs (their reasoning after first appearance).

4-6. Bacterial composition analyses outlining was low in Your case, why not used more data to enhance that in species level. That part being lower that in other studies. Some used phrases needs some revision to enhance English quality.

7. Hypothesis and clear aims are missing as well as novelty

8. Check that if your statements has not been backed up with reasoning why were NH4 oxidation and anaerobic ammonia oxidation as extensively studied field not together chosen for research, compared to other studies the novelty to be added.

9. Opportunities for reutilization of ammonia from wastewater was not well obtained, for a period of operation stabilities have not been enough, elaborate. Your nitrogen results could be enabled from other researches.

10. Only few studies were selected for inclusion in the manuscript as comparative study, could more studies be included and most novel studies involved?

 11. The choice of other researches for making research comparisons needs to be justified.

12. Aims should be followed in results and conclusion sections answering to stated aims.

13. Check spacing addition after units and numerals

13. Literature has shown different environmental friendly and other treatments to be solved for more economical way, which could be shown:https://doi.org/10.1016/j.scitotenv.2021.149133, https://doi.org/10.1080/09593330.2013.874492, https://doi.org/10.1016/S1001-0742(10)60523-2, https://doi.org/10.1080/09593330.2020.1721566, https://doi.org/10.1080/09593330.2014.941946

14. Reference style should be unified some places used different style.

15. The  influence of the type of AOB-denitrification cooperation in presence of could have been determined. What influence did AOB have and what denitrifiers, what other parameters were affecting system?

16. Copper, iron also significantly influences the physiology of these MOB and AOM due to the abundance of enzymes in the energy gain pathway, which are regarded as iron-containing proteins- Give the levels most efficient regarding to metal content.

17. Aims and hypothesis could be given. Why Cu and Fe only were studied, what is the basis of such selection?

Author Response

Comments from reviwer: #3 (Comments to the Author):

The present study investigated the the metabolic diversity of ammonia and methane oxidation related to metal compounds and the effect of various concentrations of metal compounds on MOB and AOM, which are evolutionally connected, enriched with different functional groups of bacteria.

Present review is short introduction to N and C cycle. Only one table has been added, some figures could be necessary to show distinct process differences. Nitrogen removal rates could be added as well.

This paper presents the latest achievements on the nitrogen removal and the utilization of biodegradable organic matter to provide methane and ammonia for AOB, Comammox whereas anammox process contributing important nitrogen cycle driver in marines has not been outlined. Optimal condition for AOB and MOB process could be given to be reached. Processes (ammonia oxidation to nitrate and nitrite) during an anaerobic-anoxic operation in a system was not outlined achieved.

Re) We thank the valuable comments. According to the comment, we updated the manuscript by describing other ammonia oxidation processes, including anammox, heterotrophic nitrification, and fungal nitrification. Of course, these oxidation processes are also crucial in natural and controlled conditions. However, compared to the canonical aerobic AOM, completely different physiological pathways participate in the oxidation process under other environmental conditions. Therefore in this review, we focused on canonical aerobic ammonia oxidation, carried out in aerobic conditions, and it could be directly compared to the aerobic methane oxidation pathway, which is physiologically similar. In aerobic ammonia and methane oxidation, Cu and Fe are critically important for the growth and activity of AOM and MOB. So, we also summarized the impact of various concentrations of metal compounds on the physiology of MOB and AOM. Then it could give a fundamental strategy to control MOB and AOM in diverse ecosystems because they are significantly related to climate change, eutrophication, and the remediation of contaminated sites for detoxifying pollutants. We would appreciate if you relook at the updated manuscript, providing the detail understanding of the main objectives. We also modified the title and explain various parts of the manuscript to clearly describe the main purpose of the review study.

Few remarks can be made on MS quality:

  1. “The oxidation of ammonia initiates nitrification by ammonia-oxidizing microorganisms (AOM), which is mediated by three distinct groups of aerobic autotrophic ammonia oxidizers: i) ammonia-oxidizing bacteria (AOB), ii) ammonia oxidizing-archaea (AOA), and iii) complete ammonia-oxidizing bacteria (Comammox) “ The process sequence to achieve TN after effluent treatment through AOB and anammox process seems complicated and is not sufficiently outlined, whereas also in real application many things have been done lately in these process applications. Complicated control of metals and dissolved oxygen (for MOB) and the instability of ammonia feeding at right ratio in bioreactors treating high NH4+ -N wastewater make complications in practice need to be outlined. Elaborate

    -Out as total studies metal toxicity to AOM, Comamox and denitrification is also extensively studied. Why deni and anammox was not studied thoroughly. There is bad wording in some sentences
    Re) Thank you for the comments. Similar to the answer we have already described above. We modified various points in the MS according to your suggestions by highlighting the main objective of the review study, so please check the updated parts.

  1. Work showed a amount of ammonia to be recycled by the use of Cyt, results through Comammox, which efficiencies are needed to give – Give standard deviation of your resultant pollutants’ removals, mean, max and min in separate parts available in literature.
    Re) Thank you for your comment. And we added specific ranges of the metal concentrations for specific strains in some parts of the revised MS. However, as we described in the MS, the physiological mechanism of enzymes and the growth of the microbes is mainly affected by free metal compounds, not the total metal concentrations. Furthermore, several previous studies we cited in the MS do not have clear information about the complexability between the metal and other compounds; therefore, it is not easy to identify the concentration of free metals. Nevertheless, we tried to add the concentration information as much as possible, so please consider the updated version of the MS.

  2. Needs descriptions about the exact metal parameters observed and given are to be toxic and why instabilities in current metals at their exact concentrations over how short time occurrs (their reasoning after first appearance).
    Re) As we answered above, we added specific ranges of the metal concentrations for specific strains in some parts of the revised MS. It is impossible to provide proper time, as it depends on many other environmental factors, as well as there is no exact data or results available regarding the inhibition time induced by the metals in various ecosystems.

  3. Bacterial composition analyses outlining was low in Your case, why not used more data to enhance that in species level. That part being lower that in other studies. Some used phrases needs some revision to enhance English quality.
    Re) Thank you for the suggestion. To get fundamental information and keep the main objective of the study in mind, we tried to explain the common feature of ammonia and methane oxidizers involved with copper and iron metals, while avoiding strain or species-specific studies. However, we have edited our manuscript according to the reviewer's suggestion. Please have a look at the modified manuscript.

  4. Hypothesis and clear aims are missing as well as novelty.
    Re) Thank you for the comment. We have modified the abstract and conclusion by describing and highlighting the aims of the study. Please understand and consider it.

  1. Check that if your statements has not been backed up with reasoning why were NH4 oxidation and anaerobic ammonia oxidation as extensively studied field not together chosen for research, compared to other studies the novelty to be added.
    Re) Similar comments are above, and we have already answered it. Thanks again.

  2. Opportunities for reutilization of ammonia from wastewater was not well obtained, for a period of operation stabilities have not been enough, elaborate. Your nitrogen results could be enabled from other researches.
    Re) We have changed the title and improved various sections of the MS to make it more understandable. We have clearly mentioned the aims and applications of the study. We did not concentrate on any particular application that was outside the purview of this MS, such as the "reutilization of ammonia from wastewater," as the reviewer suggested. However, as suggested by the reviewer, we thoroughly described the aims and applications in the revised MS.  

  3. Only few studies were selected for inclusion in the manuscript as comparative study, could more studies be included and most novel studies involved?
    Re) Thank you for the comment. We have tried to give a comparative analysis of various recent studies by keeping in view the scope of MS. However, we cited more than 100 references in the manuscript. Since we had to concentrate on the main purpose of the review study, we narrowed the range of subjects. Please understand and consider it.

  1. The choice of other researches for making research comparisons needs to be justified.
    Re) Many thanks for the comment. We provided a comparison of other studies carried out on the interactions of microbes and metals. However, by focusing on the objectives of our MS, we modified it to explain the applications and comparisons clearly.

  1. Aims should be followed in results and conclusion sections answering to stated aims.
    Re) Thank you for the comment. We have modified the conclusion according to your suggestions.

  2. Check spacing addition after units and numerals
    Re) Thanks for the comment. Checked and corrected.

  3. Literature has shown different environmental friendly and other treatments to be solved for more economical way, which could be shown:https://doi.org/10.1016/j.scitotenv.2021.149133, https://doi.org/10.1080/09593330.2013.874492, https://doi.org/10.1016/S1001-0742(10)60523-2, https://doi.org/10.1080/09593330.2020.1721566, https://doi.org/10.1080/09593330.2014.941946
    Re) Thank you for the suggestions. We have cited one literature in the updated manuscript.

  4. Reference style should be unified some places used different style.
    Re) Thanks and corrected it.

  5. The influence of the type of AOB-denitrification cooperation in presence of could have been determined. What influence did AOB have and what denitrifiers, what other parameters were affecting system?
    Re) Many thanks for the comment. As we described above, our main purpose of the review study is a bit different from what you think. So, to be more precise, we have changed the title and many parts of the manuscript. Our main purpose is not to summarize the nitrogen removal process in the WWTPs. Please consider it.

  1. Copper, iron also significantly influences the physiology of these MOB and AOM due to the abundance of enzymes in the energy gain pathway, which are regarded as iron-containing proteins- Give the levels most efficient regarding to metal content.
    Re) The section titled ‘Copper and iron in the ammonia-oxidizing pathway’ contain information on the influence of metals on various types of AOM. As we explained in the manuscript, diverse physiological pathways for the ammonia oxidation of various ammonia oxidizers have different impacts on the metal compounds. Therefore, we also tried to add the metal concentration information as much as possible, although, as already explained, the physiological mechanism of enzymes and the growth of the microbes are mainly affected by free metal compounds, not the total metal concentrations.

16. Aims and hypothesis could be given. Why Cu and Fe only were studied, what is the basis of such selection?

Re) Thank you for the comment. As reviewers pointed out, we have highlighted the academic contribution of the manuscript in the abstract and conclusion section. Please check the updated points. And also, we highlighted the significance of the review manuscript. Copper and iron are the key metal compounds in ammonia and methane oxidation, and it has been well addressed in the MS. Interactions between microbes and other trace metals such as zinc (Zn), cobalt (Co), molybdenum (Mo), selenium (Se), manganese (Mn) is also essential in the growth and activity of microbes. These metals contribute as key or co-factors in nitrite oxidation and CH3OH oxidation to CHOOH via CH2 We described these points in the manuscript (highlighted in blue) and Figure 1. However, as we changed the title and highlighted it in the manuscript, we would like to focus on the key metal compounds, copper and iron, effect on aerobic ammonia oxidation. Because knowing how AOM and MOB are affected by these metal compounds, is critical for understanding and controlling the global C and N cycles.

Round 2

Reviewer 2 Report

The author has revised the manuscript according to the comments. I think the revised manuscript can be accepted for the publication.